# RTFS-Net: Recurrent Time-Frequency Modelling for Efficient Audio-Visual Speech Separation

**Samuel Pegg**[1], **Kai Li**[1,*], **Xiaolin Hu**[1,2,3,†]

axh.2020@tsinghua.org.cn, lk21@mails.tsinghua.edu.cn, xlhu@tsinghua.edu.cn

1. Department of Computer Science and Technology, Institute for AI, BNRist,
Tsinghua University, Beijing 100084, China
2. Tsinghua Laboratory of Brain and Intelligence (THBI),
IDG/McGovern Institute for Brain Research, Tsinghua University, Beijing 100084, China
3. Chinese Institute for Brain Research (CIBR), Beijing 100010, China

## Abstract

Audio-visual speech separation methods aim to integrate different modalities to generate high-quality separated speech, thereby enhancing the performance of downstream tasks such as speech recognition. Most existing state-of-the-art (SOTA) models operate in the time domain. However, their overly simplistic approach to modeling acoustic features often necessitates larger and more computationally intensive models in order to achieve SOTA performance. In this paper, we present a novel time-frequency domain audio-visual speech separation method: Recurrent Time-Frequency Separation Network (RTFS-Net), which applies its algorithms on the complex time-frequency bins yielded by the Short-Time Fourier Transform. We model and capture the time and frequency dimensions of the audio independently using a multi-layered RNN along each dimension. Furthermore, we introduce a unique attention-based fusion technique for the efficient integration of audio and visual information, and a new mask separation approach that takes advantage of the intrinsic spectral nature of the acoustic features for a clearer separation. RTFS-Net outperforms the prior SOTA method in both **inference speed** and **separation quality** while reducing the number of parameters by **90%** and MACs by **83%**. This is the first time-frequency domain audio-visual speech separation method to outperform all contemporary time-domain counterparts.

## 1 Introduction

The 'cocktail party problem' (Bronkhorst, 2000; Haykin & Chen, 2005; Cherry, 2005) describes our ability to focus on a single speaker's voice amidst numerous overlapping voices in a noisy environment. While humans effortlessly tackle this problem, replicating this ability in machines remains a longstanding challenge in the field of signal-processing. Audio-only Speech Separation (AOSS) methods (Luo & Mesgarani, 2019; Luo et al., 2020; Subakan et al., 2021), solely utilizing the mixed-speaker audio signal, face limitations in scenarios with strong background noise, reverberation, or heavy voice overlap. To overcome these issues, researchers turned to a multi-modal approach: Audio-visual Speech Separation (AVSS) (Gao & Grauman, 2021; Lee et al., 2021; Li et al., 2022). AVSS methods integrate additional visual cues into the paradigm and can be generally split into two main classifications: Time-domain (T-domain) and Time-Frequency domain (TF-domain) methods, each with their own benefits and challenges.

T-domain methods (Wu et al., 2019; Li et al., 2022; Martel et al., 2023; Lin et al., 2023) work on the long, uncompressed and high-dimensional audio features returned by the 1D convolutional encoder design proposed by Luo & Mesgarani (2018). This approach facilitates fine-grained, high quality audio separation, but the high parameter count and computational complexity leads to extended training periods, intensive GPU usage and slow inference speeds. On the other hand, TF-domain

---

*Made significant contributions. † Corresponding author.

methods (Afouras et al., 2018a; Alfouras et al., 2018; Gao & Grauman, 2021) apply their algorithms on the complex 2D representation yielded by the Short-Time Fourier Transform (STFT). Typically, the STFT uses large windows and hop lengths to compress the data, resulting in more computationally efficient separation methods. However, from a historical perspective, all TF-domain methods have been substantially outperformed by T-domain methods. Based on our research, this gap in performance stems from three critical factors.

Firstly, while some attempts have been made (Afouras et al., 2020; Lee et al., 2021) to model amplitude and phase separately, no TF-domain AVSS methods explore the independent and tailored modelling of the two acoustic dimensions (time and frequency) in order to exploit this domain's advantage over T-domain methods. Indeed, recent research by Luo & Yu (2023) and Wang et al. (2023) in the AOSS domain have capitalized on this advantage and outperformed their T-domain counterparts by large margins. However, their usage of large LSTM (Hochreiter & Schmidhuber, 1997) and transformer (Vaswani et al., 2017) architectures leads to a heavy computational burden, making them an unattractive solution in the AVSS space. Secondly, while existing AVSS studies (Li et al., 2022; Lin et al., 2023) have explored various audio-visual fusion strategies, they neglect using visual features from multiple receptive fields to increase model performance. This type of visual information serves as important ā priōrī knowledge crucial for accurately extracting the target speaker's voice. Thirdly, TF-domain AVSS studies (Afouras et al., 2020; Lee et al., 2021; Gao & Grauman, 2021) often overlook the underlying complex nature of the features, and hence lose critical amplitude and phase information when extracting the target speaker's voice from the audio mixture. This degrades the reconstruction performance of the inverse STFT (iSTFT) and leads to poorer model performance.

In this work, we propose a novel TF-domain AVSS method: *Recursive Time-Frequency Separation Network* (RTFS-Net, Figure 1) that provides a computationally efficient solution to the cocktail party problem using an STFT based audio encoder, a high-fidelity pretrained video encoder and a series of recursive RTFS Blocks to perform target speaker extraction. Our contributions are threefold:

1. Our *RTFS Blocks* tackle the first issue by projecting the features to a compressed subspace. There, we explicitly model both acoustic dimensions individually, then in tandem before subsequently applying an attentional mechanism (TF-AR) to restore the dimensions with minimal information loss, allowing us to reap the benefits of independent time-frequency processing without bearing a substantial computational cost.

2. Our *Cross-dimensional Attention Fusion* (CAF) Block provides a low parameter, computationally efficient solution to the second issue by aggregating the multi-modal information through a multi-head attention strategy in order to optimally fuse the visual cues of the target speaker into the audio features, facilitating high quality separation.

3. Our *Spectral Source Separation* ($S^3$) Block addresses the third issue by explicitly reconstructing the plural features of the target speaker, achieving a higher quality separation without increasing computational cost.

We conducted comprehensive experimental evaluations on three widely used datasets: LRS2 (Afouras et al., 2018a), LRS3 (Afouras et al., 2018b) and VoxCeleb2 (Chung et al., 2018), to demonstrate the value of each contribution. To the best of our knowledge, RTFS-Net is the first TF-domain AVSS method to outperform all contemporary T-domain methods, achieving this while also exhibiting a clear advantage by reducing computational complexity by 83% and reducing the parameter count by 90%. We additionally provide a Web page where sample results can be listened to, and our code will be open-sourced after publication for reproducibility purposes.

## 2 RELATED WORK

**Audio-only speech separation**. Modern deep learning advances introduced neural networks for speaker-agnostic speech separation, with Conv-TasNet (Luo & Mesgarani, 2019) and DualPathRNN (Luo et al., 2020) making significant T-domain breakthroughs. However, T-domain methods (Luo & Mesgarani, 2019; Luo et al., 2020) often exhibit a marked performance degradation in reverberant conditions, attributed to their neglect of explicit frequency-domain modeling. Consequently, recent research focuses on high-performance speech separation models in the TF-domain. For instance, TFPSNet (Yang et al., 2022) incorporates the transformer from DPTNet (Chen et al., 2020) to

assimilate spectral-temporal information. TF-GridNet (Wang et al., 2023) extends this by introducing a cross-frame self-attention module in order to achieve SOTA performance on the WSJ0-2mix (Hershey et al., 2016). However, the inclusion of many LSTM and Transformer layers results in extremely high computational complexity, leading to increased training times and GPU memory requirements. Lightweight models like A-FRCNN (Hu et al., 2021) and TDANet (Li et al., 2023) strike a balance between performance and efficiency by using an encoder-decoder paradigm with recurrent connections and top-down attention. However, their separation prowess in noisy scenarios remains suboptimal compared to multi-modal approaches.

**Audio-visual speech separation**. Conventional AVSS statistical methods relied on knowledge-driven modeling (Loizou, 2013; Wang & Brown, 2006). Early deep-learning explorations into AVSS mainly occurred in the TF-domain, focusing on tasks like amplitude and phase reconstruction for target speakers (Afouras et al., 2018a; Alfouras et al., 2018). With the advent of AV-ConvTasNet (Wu et al., 2019), it became a prevailing belief that T-domain AVSS methods consistently outperformed TF-domain methods. Notably, CTCNet (Li et al., 2022), inspired by thalamic brain structures, introduced a unique multiple-fusion approach. Leveraging multiscale contexts from visual and auditory cues, this module greatly enhanced the spatial scale of fused features, thus improving the model's separation capacity. However, as mentioned previously, T-domain methods come with a higher computational load. TF-domain AVSS techniques often use larger windows and hop sizes, which curtails computational complexity. Nevertheless, their full potential is yet to be realized. The recent Visualvoice model (Gao & Grauman, 2021) combined a multi-task learning framework for both AVSS and cross-modal speaker embedding, incorporating facial expressions, lip movements, and audio cues. Despite its touted efficiency, Visualvoice lacks robust modeling, and it lags behind many modern T-domain methods.

# 3 METHODS

Expanding on prior SOTA methods (Wu et al., 2019; Li et al., 2022), we present our AVSS pipeline, illustrated in Figure 1. The mono-aural mixed-speech audio signal, $\boldsymbol{x} \in \mathbb{R}^{1 \times L_\mathrm{a}}$, in conjunction with the video frames capturing the target speaker's lip movements, $\boldsymbol{y} \in \mathbb{R}^{1 \times L_\mathrm{v} \times H \times W}$, are used as inputs to RTFS-Net in order to derive the target speaker's estimated audio signal, $\hat{\boldsymbol{s}} \in \mathbb{R}^{1 \times L_\mathrm{a}}$. In this context, $L_\mathrm{a}$ and $L_\mathrm{v}$ signify the durations of the audio and video inputs, while $H$ and $W$ correspond to the dimensions of the single-channel (grey-scale) video frames.

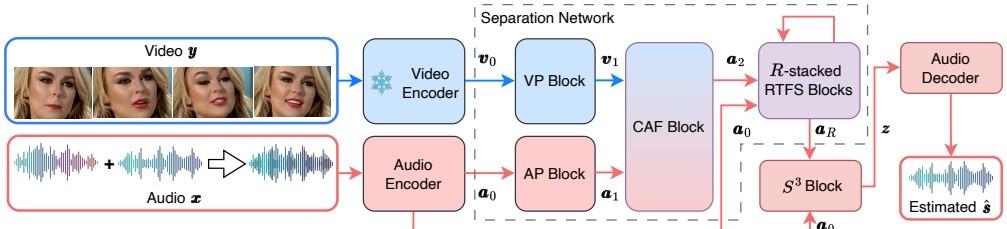

Figure 1: The overall pipeline of RTFS-Net. The red and blue solid lines signify the flow directions of auditory and visual features respectively. The snowflake indicates the weights are frozen and the component is not involved in training.

Firstly, the audio and video encoders extract auditory $\boldsymbol{a}_0$ and visual $\boldsymbol{v}_0$ features. These serve as the inputs for our separation network, which fuses these features and extracts the salient multimodal features, $\boldsymbol{a}_R$. Next, our Spectral Source Separation ($S^3$) method is applied to separate the target speaker's audio $\boldsymbol{z}$ from the encoded audio signal $\boldsymbol{a}_0$ using $\boldsymbol{a}_R$. Finally, the target speaker's estimated audio feature map $\boldsymbol{z}$ is decoded into the estimated audio stream $\hat{\boldsymbol{s}}$ and compared with the ground-truth signal $\boldsymbol{s}$ for training.

## 3.1 ENCODERS

Our encoders distill relevant features from both audio and visual inputs. For the video encoder $E_\mathrm{v}(\cdot)$, we employ the CTCNet-Lip (Li et al., 2022) pretrained network to extract the visual features

$\boldsymbol{v}_0$ of the target speaker,

$$\boldsymbol{v}_0 = E_{\mathrm{v}}(\boldsymbol{y}), \quad \boldsymbol{v}_0 \in \mathbb{R}^{C_{\mathrm{v}} \times T_{\mathrm{v}}}. \tag{1}$$

For the audio encoder, we firstly define $\boldsymbol{\alpha}$ as the complex-valued hybrid TF-domain bins obtained using the STFT on $\boldsymbol{x}$. For a mixed audio of $n_{\mathrm{spk}}$ speakers, we define $\boldsymbol{s}_i$ as the speech of speaker $i$ and $\boldsymbol{\epsilon}$ as the presence of some background noise, music, or other extraneous audio sources. Then,

$$\boldsymbol{\alpha}(t, f) = \boldsymbol{\epsilon}(t, f) + \sum_{i=1}^{n_{\mathrm{spk}}} \boldsymbol{s}_i(t, f), \quad \boldsymbol{\alpha}(t, f) \in \mathbb{C} \,\forall\, (t, f), \tag{2}$$

where $t \in [0, T_{\mathrm{a}}]$ is the time dimension and $f \in [0, F]$ is the frequency dimension. We concatenate the real (Re) and imaginary (Im) parts of $\boldsymbol{\alpha}$ along a new 'channels' axis, then apply a 2D convolution $E_{\mathrm{a}}(\cdot)$ with a $3 \times 3$ kernel and $C_{\mathrm{a}}$ output channels across the time and frequency dimensions to obtain the auditory embedding $\boldsymbol{a}_0$ of $\boldsymbol{x}$. Using the symbol $||$ for concatenation, we write,

$$\boldsymbol{a}_0 = E_{\mathrm{a}}\left(\mathrm{Re}(\boldsymbol{\alpha})||\mathrm{Im}(\boldsymbol{\alpha})\right), \quad \boldsymbol{a}_0 \in \mathbb{R}^{C_{\mathrm{a}} \times T_{\mathrm{a}} \times F}. \tag{3}$$

## 3.2 SEPARATION NETWORK

The core of RTFS-Net is a separation network that uses recursive units to facilitate information interaction in the two acoustic dimensions, and efficiently aggregates multimodal features using an attention-based fusion mechanism. The initial step is to preprocess the auditory $\boldsymbol{a}_0$ and visual $\boldsymbol{v}_0$ features separately in preparation for fusion. For the Visual Preprocessing (VP) Block we adopt a modified version of the TDANet Block (Li et al., 2023), see Appendix A. For the Audio Preprocessing (AP) Block we use a single RTFS Block, whose structure will be defined in Section 3.2.2. The outputs of the two preprocessing blocks are fed to our proposed CAF Block to fuse the multimedia features into a single enriched feature map (see Figure 1 and 2). This audio-visual fusion is subsequently processed with an additional $R$ stacked RTFS Blocks. Following CTCNet (Li et al., 2022), these $R$ sequential blocks share parameters (including the AP Block), which has been shown to reduce model size and increase performance, since it turns the series of blocks in to a recurrent neural architecture (see Appendix B).

### 3.2.1 CROSS-DIMENSIONAL ATTENTION FUSION BLOCK

The CAF Block (Figure 2) is a depth-wise and group-convolution based architecture designed to consume as little resources as possible while fusing the 2D visual data into the 3D audio data. It involves two separate fusion operations that we call the *attention* fusion ($\boldsymbol{f}_1$) and the *gated* fusion ($\boldsymbol{f}_2$). The attention fusion considers multiple visual sub-representation spaces to aggregate information from a wide receptive field and apply attention to the audio features. The gated fusion up-samples the visual information's time dimension, then expands the visual features into the TF-domain using $F$ gates produced from the preprocessed audio features. We use Vaswani et al. (2017)'s 'keys' and 'values' nomenclature.

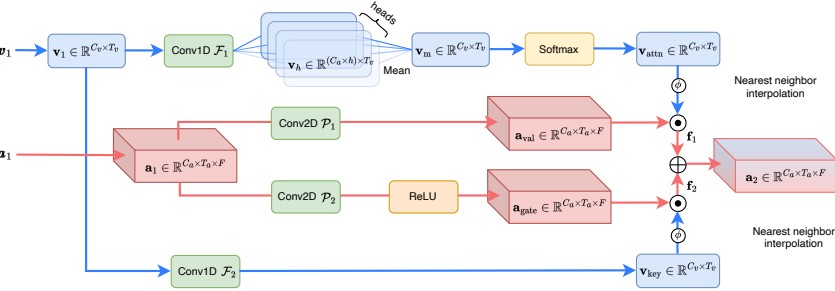

Figure 2: Structure of our CAF Block. As in Figure 1, the red and blue solid lines signify the flow directions of auditory and visual features respectively.

Firstly, let both $\mathcal{P}_1$ and $\mathcal{P}_2$ denote a depth-wise convolution with a $1 \times 1$ kernel and following global layer normalization (gLN) (Luo & Mesgarani, 2019). We generate the audio 'value' embeddings and the aforementioned 'gate' from the preprocessed audio signal $\boldsymbol{a}_1$.

$$\boldsymbol{a}_{\mathrm{val}} = \mathcal{P}_1(\boldsymbol{a}_1), \quad \boldsymbol{a}_{\mathrm{gate}} = \mathrm{ReLU}\left(\mathcal{P}_2(\boldsymbol{a}_1)\right), \qquad \boldsymbol{a}_{\mathrm{val}}, \boldsymbol{a}_{\mathrm{gate}} \in \mathbb{R}^{C_{\mathrm{a}} \times T_{\mathrm{a}} \times F}. \tag{4}$$

**Attention Fusion**. We apply a 1D group convolution $\mathcal{F}_1$ with $C_\mathrm{a}$ groups and $C_\mathrm{a} \times h$ output channels to $\boldsymbol{v}_1$, followed by a gLN layer. By chunking across the channels, we can decompose the visual features into $h$ distinct sub-feature representations, or attention 'heads', $\boldsymbol{v}_\mathrm{h}$. Next, we take the 'mean' of the $h$ heads to aggregate the information from the different sub-feature representations into $\boldsymbol{v}_\mathrm{m}$, then subsequently apply the Softmax operation in order to create a multi-head attention style set of features $\boldsymbol{v}_\mathrm{attn}$ with values between 0 and 1. To align the video frame length $T_\mathrm{v}$ with the audio's time dimension $T_\mathrm{a}$, we use nearest neighbor interpolation, $\phi$. This is written,

$$\boldsymbol{v}_h = \mathcal{F}_1(\boldsymbol{v}_1), \qquad\qquad \boldsymbol{v}_h \in \mathbb{R}^{(C_\mathrm{a} \times h) \times T_\mathrm{v}}, \tag{5}$$

$$\boldsymbol{v}_m = \mathrm{mean}(\boldsymbol{v}_h[1], \ldots, \boldsymbol{v}_h[h])), \qquad \boldsymbol{v}_m \in \mathbb{R}^{C_\mathrm{a} \times T_\mathrm{v}}, \tag{6}$$

$$\boldsymbol{v}_\mathrm{attn} = \phi(\mathrm{Softmax}(\boldsymbol{v}_m)), \qquad \boldsymbol{v}_\mathrm{attn} \in \mathbb{R}^{C_\mathrm{a} \times T_\mathrm{a}}. \tag{7}$$

Our attention mechanism is applied to each of the $F$ 'value' slices of $\boldsymbol{a}_\mathrm{val}$ with length $T_\mathrm{a}$,

$$\boldsymbol{f}_1[i] = \boldsymbol{v}_\mathrm{attn} \odot \boldsymbol{a}_\mathrm{val}[i], \quad \forall i \in \{1, \ldots, F\}, \tag{8}$$

where $\boldsymbol{f}_1[i] \in \mathbb{R}^{C_a \times T_\mathrm{a}} \; \forall i \implies \boldsymbol{f}_1 \in \mathbb{R}^{C_a \times T_\mathrm{a} \times F}$.

**Gated Fusion**. We use a 1D convolutional layer $\mathcal{F}_2$ with kernel size 1, $C_\mathrm{a}$ output channels and $C_\mathrm{a}$ groups (since $C_\mathrm{a} < C_\mathrm{v}$), followed by a gLN layer to align $C_\mathrm{v}$ with $C_\mathrm{a}$. Next, we again use interpolation $\phi$ to align $T_\mathrm{v}$ with $T_\mathrm{a}$ and generate the visual 'key' embeddings.

$$\boldsymbol{v}_\mathrm{key} = \phi\left(\mathcal{F}_2(\boldsymbol{v}_1)\right), \quad \boldsymbol{v}_\mathrm{key} \in \mathbb{R}^{C_\mathrm{a} \times T_\mathrm{a}}. \tag{9}$$

Next, we utilize all $F$ of the $T_\mathrm{a}$-dimensional slices of $\boldsymbol{a}_\mathrm{gate}$ as unique gates to comprehensively expand the visual information into the TF-domain,

$$\boldsymbol{f}_2[i] = \boldsymbol{a}_\mathrm{gate}[i] \odot \boldsymbol{v}_\mathrm{key}, \quad \forall i \in \{1, \ldots, F\}, \tag{10}$$

where $\boldsymbol{f}_2[i] \in \mathbb{R}^{C_\mathrm{a} \times T_\mathrm{a}} \; \forall i \implies \boldsymbol{f}_2 \in \mathbb{R}^{C_\mathrm{a} \times T_\mathrm{a} \times F}$.

**CAF Block**. Finally, we sum the two fused features together. We can denote our CAF Block, $\Phi$, as:

$$\boldsymbol{a}_2 = \Phi(\boldsymbol{a}_1, \boldsymbol{v}_1) = \boldsymbol{f}_1 + \boldsymbol{f}_2, \quad \boldsymbol{a}_2 \in \mathbb{R}^{C_\mathrm{a} \times T_\mathrm{a} \times F}. \tag{11}$$

### 3.2.2   RTFS BLOCKS

Compared to previous TF-domain AVSS methods (Afouras et al., 2020; Gao & Grauman, 2021; Alfouras et al., 2018; Lee et al., 2021), our RTFS Blocks use a dual-path architecture to explicitly model audio in both acoustic dimensions to improve the separation quality, as shown in Figure 3. We denote the auditory features input into the RTFS Block as $\boldsymbol{A}$. Given our recurrent structure, $\boldsymbol{A}$ represents either $\boldsymbol{a}_0$ (input to AP Block) or the output from the previous RTFS Block with a skip connection: $\boldsymbol{a}_j + \boldsymbol{a}_0$ for $j \in \{1, ..., R\}$. Note that in Figure 1 the residual connection is not shown for simplicity. Our RTFS Block processes auditory features in the four steps discussed below.

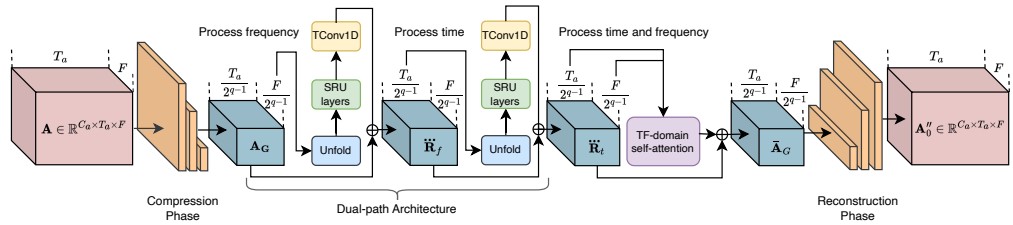

Figure 3: RTFS Block design. After compressing the data to a more efficient size, we process first the frequency dimension, then the time dimension, then both dimensions in tandem using TF-domain self-attention to capture inter-dependencies. We then carefully restore the data to its original dimensions using our Temporal-Frequency Attention Reconstruction units.

**Compression of time and frequency resolution**. We use a 2D convolution with a $1 \times 1$ kernel to convert $\boldsymbol{A}$ to a smaller channel dimension $D < C_\mathrm{a}$. This means we can effectively employ a larger $C_\mathrm{a}$ value for detailed fusion (CAF) and separation ($S^3$), while maintaining a lightweight and efficient block design. Similar to Li et al. (2023), in the compression phase we employ $q$ stacked 2D depth-wise convolutional layers with $4 \times 4$ kernels and stride 2, see Appendix C for the effects

of different $q$ values. The resultant multi-scale set with varying temporal and frequency resolutions can be denoted $\{\boldsymbol{A}_i | i \in \{0, q-1\}\}$, where $\boldsymbol{A}_i \in \mathbb{R}^{D \times \frac{T_a}{2^i} \times \frac{F}{2^i}}$. We use adaptive average pooling $p$ to compress each member of the set to the dimensions of the smallest member, $\boldsymbol{A}_{q-1}$, then sum to obtain the compressed global features $\boldsymbol{A}_G$. This is written:

$$\boldsymbol{A}_G = \sum_{i=0}^{q-1} p(\boldsymbol{A}_i), \quad \boldsymbol{A}_G \in \mathbb{R}^{D \times \frac{T_a}{2^{q-1}} \times \frac{F}{2^{q-1}}}. \tag{12}$$

**Dual-path architecture**. The Dual-path RNN architecture has been extensively deployed in AOSS tasks (Luo et al., 2020; Chen et al., 2020; Wang et al., 2023). However, their usage of large LSTMs leads to high parameter counts and an elevated computational complexity. In natural language processing, Simple Recurrent Units (SRU (Lei et al., 2018)) were introduced to replace LSTMs by processing most of the operations in parallel, speeding up model training and inference time. Inspired by this, we adopt SRUs for the AVSS task, see Appendix D for a detailed analysis between different recurrent architectures.

As seen in Figure 3, we first process the *frequency* dimension, then the *time* dimension. As with all dual-path methods, the SRUs are applied across each slice in the *time* dimension to process the *frequency* dimension. Similar to (Wang et al., 2023), we unfold[1] the features by zero-padding the frequency dimension of $\boldsymbol{A}_G$, then unfolding with kernel size 8 and stride 1,

$$\dot{\boldsymbol{R}}_f = \left[ \text{Unfold}(\boldsymbol{A}_G[:, t]), \ t \in \left\{0, \ldots, T_a/2^{q-1}\right\} \right] \in \mathbb{R}^{8D \times \frac{T_a}{2^{q-1}} \times F'}, \tag{13}$$

where $F'$ is the resulting padded and unfolded frequency dimension. Note that while this process results in high input dimensions for the SRUs, which significantly increases computational complexity, it is essential for achieving outstanding separation performance. This is why it is so important that we first compress the time-frequency resolution. After unfolding, layer normalization is applied in the channel dimension, and then a bidirectional, 4 layer SRU with hidden size $h_a$ is applied,

$$\ddot{\boldsymbol{R}}_f = \left[ \text{SRU}(\dot{\boldsymbol{R}}_f[:, t]), \ t \in \left\{0, \ldots, T_a/2^{q-1}\right\} \right] \in \mathbb{R}^{2h_a \times \frac{T_a}{2^{q-1}} \times F'}. \tag{14}$$

A transposed convolution $\mathcal{T}$ with kernel 8 and stride 1 is used to restore the unfolded dimensions,

$$\dddot{\boldsymbol{R}}_f = \mathcal{T}\left(\ddot{\boldsymbol{R}}_f\right) + \boldsymbol{A}_G, \quad \dddot{\boldsymbol{R}}_f \in \mathbb{R}^{D \times \frac{T_a}{2^{q-1}} \times \frac{F}{2^{q-1}}}. \tag{15}$$

We next process the *time* dimension using the same method, and then finally apply Wang et al. (2023)'s TF-domain self-attention network, denoted $\text{Attn}$. These two steps are expressed below as:

$$\dddot{\boldsymbol{R}}_t = \mathcal{T}\left(\ddot{\boldsymbol{R}}_t\right) + \dddot{\boldsymbol{R}}_f, \quad \bar{\boldsymbol{A}}_G = \text{Attn}(\dddot{\boldsymbol{R}}_t) + \dddot{\boldsymbol{R}}_t, \qquad \dddot{\boldsymbol{R}}_t, \bar{\boldsymbol{A}}_G \in \mathbb{R}^{D \times \frac{T_a}{2^{q-1}} \times \frac{F}{2^{q-1}}}. \tag{16}$$

**Reconstruction of time and frequency resolution**. Reconstruction of high-quality temporal and frequency features presents a formidable challenge. Delving into the underlying causes, the reconstruction process often relies on interpolation or transposed convolutions for up-sampling, resulting in the emergence of checkerboard artifacts in the reconstructed outputs. To solve this problem, we propose the Temporal-Frequency Attention Reconstruction (TF-AR) unit, denoted $I(\cdot, \cdot)$. This unit prioritizes the reconstruction of key features by exploiting an attention mechanism, thus reducing information loss. For two tensors $\boldsymbol{m}$ and $\boldsymbol{n}$, we define the TF-AR unit as:

$$I(\boldsymbol{m}, \boldsymbol{n}) = \phi\left(\sigma\left(W_1\left(\boldsymbol{n}\right)\right)\right) \odot W_2\left(\boldsymbol{m}\right) + \phi\left(W_3\left(\boldsymbol{n}\right)\right), \tag{17}$$

where $W_1(\cdot)$, $W_2(\cdot)$ and $W_3(\cdot)$ denote 2D depth-wise convolutions with $4 \times 4$ kernels followed by a gLN layer. We use the notation $\sigma$ for the sigmoid function, $\odot$ for element-wise multiplication and $\phi$ for nearest neighbour interpolation (up-sampling). To conduct the reconstruction, we firstly use $q$ TF-AR units to fuse $\bar{\boldsymbol{A}}_G$ with every element of $\boldsymbol{A}_i$,

$$\boldsymbol{A}_i' = I(\boldsymbol{A}_i, \bar{\boldsymbol{A}}_G), \quad \boldsymbol{A}_i' \in \mathbb{R}^{D \times \frac{T_a}{2^i} \times \frac{F}{2^i}} \ \forall i \in \{0, \ldots, q-1\}. \tag{18}$$

Next, the multi-scale features are continuously up-sampled and aggregated using $q - 1$ additional TF-AR units to obtain the finest-grained auditory features, $\boldsymbol{A}_0'' \in \mathbb{R}^{D \times T_a \times F}$. A residual connection to $\{\boldsymbol{A}_i | i \in \{0, q-2\}\}$ is crucial, and creates a U-Net (Ronneberger et al., 2015) style structure:

$$\boldsymbol{A}_{q-1-i}'' = I(\boldsymbol{A}_{q-1-i}', \boldsymbol{A}_{q-i}') + \boldsymbol{A}_{q-1-i}, \quad \forall i \in \{0, \ldots, q-2\}. \tag{19}$$

Finally, $\boldsymbol{A}_0''$ is converted back to $C_a$ channels using a 2D convolution with a $1 \times 1$ kernel and a residual connection to the input of the RTFS Block is added. The output features are used as the audio input for the CAF Block after the AP Block, and as the input to the next RTFS Block during the repeated $R$ stacked RTFS Blocks stage of the separation network.

---

[1] https://pytorch.org/docs/stable/generated/torch.nn.Unfold.html

## 3.3 SPECTRAL SOURCE SEPARATION

The majority of existing T-domain AVSS methods (Wu et al., 2019; Li et al., 2022; Martel et al., 2023) generate a mask $\boldsymbol{m}$ from the refined features $\boldsymbol{a}_R$, then use element-wise multiplication $\odot$ between the encoded audio mixture $\boldsymbol{a}_0$ and the mask in order to obtain the target speaker's separated speech $\boldsymbol{z}$. This is written,

$$\boldsymbol{z} = \boldsymbol{m} \odot \boldsymbol{a}_0. \tag{20}$$

Some TF-domain AVSS methods (Afouras et al., 2018a; Gao & Grauman, 2021; Lee et al., 2021) directly apply this approach to the TF-domain setting without modification, while other methods (Alfouras et al., 2018; Owens & Efros, 2018) choose not to use masks at all, and directly passes the output of the separation network $\boldsymbol{a}_R$ to the decoder. However, we found both of these TF-domain methods for target speaker extraction to be suboptimal. We need to pay attention to the underlying complex nature of the audio features produced by the STFT in order to obtain a clearer distinction. This leads us to introduce our $S^3$ Block, which utilizes a high-dimensional application of the multiplication of complex numbers, Equation 21, to better preserve important acoustic properties during the speaker extraction process, see Section 5.2 and Appendix E.

$$(a + bi)(c + di) = ac - bd + i(ad + bc). \tag{21}$$

Firstly, a mask $\boldsymbol{m}$ is generated from $\boldsymbol{a}_R$ using a 2D convolution $\mathcal{M}$ with a $1 \times 1$ kernel,

$$\boldsymbol{m} = \mathrm{ReLU}\left(\mathcal{M}\left(\mathrm{PReLU}(\boldsymbol{a}_R)\right)\right), \quad \boldsymbol{m} \in \mathbb{R}^{C_{\mathrm{a}} \times T_{\mathrm{a}} \times F}. \tag{22}$$

Without loss of generality, we choose the top half of the channels as the real part, and the bottom half of the channels as the imaginary part. We hence define,

$$\boldsymbol{m}^{\mathrm{r}} = \boldsymbol{m}\left[0 : C_{\mathrm{a}}/2 - 1\right], \qquad\qquad \boldsymbol{E}^{\mathrm{r}} = \boldsymbol{a}_0\left[0 : C_{\mathrm{a}}/2 - 1\right], \tag{23}$$

$$\boldsymbol{m}^{\mathrm{i}} = \boldsymbol{m}\left[C_{\mathrm{a}}/2 : C_{\mathrm{a}}\right], \qquad\qquad \boldsymbol{E}^{\mathrm{i}} = \boldsymbol{a}_0\left[C_{\mathrm{a}}/2 : C_{\mathrm{a}}\right]. \tag{24}$$

Next, with $||$ denoting concatenation along the channel axis, we apply Equation 21 and calculate:

$$\boldsymbol{z}^{\mathrm{r}} = \boldsymbol{m}^{\mathrm{r}} \odot \boldsymbol{E}^{\mathrm{r}} - \boldsymbol{m}^{\mathrm{i}} \odot \boldsymbol{E}^{\mathrm{i}}, \tag{25}$$

$$\boldsymbol{z}^{\mathrm{i}} = \boldsymbol{m}^{\mathrm{r}} \odot \boldsymbol{E}^{\mathrm{i}} + \boldsymbol{m}^{\mathrm{i}} \odot \boldsymbol{E}^{\mathrm{r}}, \tag{26}$$

$$\boldsymbol{z} = (\boldsymbol{z}^{\mathrm{r}} \,||\, \boldsymbol{z}^{\mathrm{i}}), \qquad \boldsymbol{z} \in \mathbb{R}^{C_{\mathrm{a}} \times T_{\mathrm{a}} \times F}, \tag{27}$$

to obtain $\boldsymbol{z}$, the target speaker's separated encoded audio features.

## 3.4 DECODER

The decoder $D(\cdot)$ takes the separated target speaker's audio features $\boldsymbol{z}$ and reconstructs the estimated waveform $\hat{\boldsymbol{s}} = D(\boldsymbol{z})$, where $\hat{\boldsymbol{s}} \in \mathbb{R}^{1 \times L_{\mathrm{a}}}$. Specifically, $\boldsymbol{z}$ is passed through a transposed 2D convolution with a $3 \times 3$ kernel and 2 output channels. Mirroring the encoder, we take the first channel as the real part and the second channel as the imaginary part and form a complex tensor. This tensor is passed to the iSTFT to recover the estimated target speaker audio.

## 4 EXPERIMENTAL SETUP

**Datasets**. We utilized the same AVSS datasets as previous works (Gao & Grauman, 2021; Li et al., 2022) in the field in order to create a fair comparison of performance: LRS2-2Mix (Afouras et al., 2018a), LRS3-2Mix (Afouras et al., 2018b) and VoxCeleb2-2Mix (Chung et al., 2018). The models were trained and tested on two second 25fps video clips with an audio sampling rate of 16 kHz. This equates to 32,000 audio frames and 50 video frames, see Appendix F for more details.

**Evaluation**. Following recent literature (Li et al., 2022), SI-SNRi and SDRi were used to evaluate the quality of the separated speeches, see Appendix G. We also provided PESQ (Rix et al., 2001) for reference. For these metrics, a higher value indicates better performance. The parameter counts displayed in the results tables are the number of trainable parameters, excluding the pretrained video model. Likewise, the number of Multiply-ACcumulate (MAC) operations indicates the MACs used while processing two seconds of audio at 16 kHz, excluding the pretrained video network. In our main results table, we also include inference time: the time taken to process 2 seconds of audio on a NVIDIA 2080 GPU. For parameters, MACs and inference speed, a lower value is preferable and is the main focus of this work. Model hyperparameter configurations are available in Appendix H. All training settings are available in Appendix I.

## 5 RESULTS

### 5.1 COMPARISONS WITH STATE-OF-THE-ART METHODS

The goal of this research is to obtain SOTA performance while significantly reducing computational complexity and model size with a lightweight, streamlined approach. In the comprehensive comparison presented in Table 1, we evaluated RTFS-Net against a range of SOTA AVSS methods. We explored three variants of RTFS-Net, corresponding to $R = \{4, 6, 12\}$ RTFS Blocks, including the AP Block. On the LRS2-2Mix dataset, RTFS-Net-4 demonstrated an SI-SNRi of 14.1 dB, only marginally lower than the 14.3 dB achieved by the previous SOTA, CTCNet. However, this was achieved with a tenfold reduction in model parameters and an eightfold decrease in computational cost. Meanwhile, RTFS-Net-6 surpassed CTCNet on the LRS2-2Mix dataset and delivered comparable results on VoxCeleb2-2Mix, striking a good balance between performance and efficiency. RTFS-Net-12 outperformed all other techniques across all datasets, showcasing its superiority in complex environments and the robustness of our TF-domain approach. Moreover, despite utilizing 12 layers, RTFS-Net-12 still manages to reduce the computational costs of CTCNet by threefold while using only a tenth of the parameters. To our knowledge, RTFS-Net is the first AVSS method to employ fewer than 1 million parameters, and the first TF-domain model to outperform all T-domain counterparts.

| Model | LRS2-2Mix | | | LRS3-2Mix | | | VoxCeleb2-2Mix | | | Params | MACs | Time |
| | SI-SNRi | SDRi | PESQ | SI-SNRi | SDRi | PESQ | SI-SNRi | SDRi | PESQ | (M) | (G) | (ms) |
|---|---|---|---|---|---|---|---|---|---|---|---|---|
| CaffNet-C* 2021 | - | 12.5 | 1.15 | - | 12.3 | - | - | - | - | - | - | - |
| Thanh-Dat 2021 | - | 11.6 | - | - | - | - | - | - | - | - | - | - |
| AV-ConvTasnet 2019 | 12.5 | 12.8 | - | 11.2 | 11.7 | - | 9.2 | 9.8 | - | 16.5 | - | 60.3 |
| VisualVoice 2021 | 11.5 | 11.8 | 2.78 | 9.9 | 10.3 | - | 9.3 | 10.2 | - | 77.8 | - | 130.2 |
| AVLIT 2023 | 12.8 | 13.1 | 2.56 | 13.5 | 13.6 | 2.78 | 9.4 | 9.9 | 2.23 | 5.8 | 36.4 | **53.4** |
| CTCNet 2022 | 14.3 | 14.6 | **3.08** | 17.4 | 17.5 | 3.24 | 11.9 | 13.1 | **3.00** | 7.0 | 167.2 | 122.7 |
| RTFS-Net-4 | 14.1 | 14.3 | 3.02 | 15.5 | 15.6 | 3.08 | 11.5 | 12.4 | 2.94 | **0.7** | **21.9** | 57.8 |
| RTFS-Net-6 | 14.6 | 14.8 | 3.03 | 16.9 | 17.1 | 3.12 | 11.8 | 12.8 | 2.97 | **0.7** | 30.5 | 64.7 |
| RTFS-Net-12 | **14.9** | **15.1** | 3.07 | **17.5** | **17.6** | **3.25** | **12.4** | **13.6** | **3.00** | **0.7** | 56.4 | 109.9 |

Table 1: Comparison of RTFS-Net with existing AVSS methods on the LRS2-2Mix, LRS3-2Mix and VoxCeleb2-2Mix datasets. These metrics are averaged across all speakers for each test set, larger SI-SNRi and SDRi values indicate better performance. '-' indicates the results are not reported in the original paper. '*' indicates that the audio is reconstructed using the ground-truth phase.

**Important Note**: we investigated higher $R$ values in Appendix J and found that RTFS-Net-20 can outperform CTCNet by a substantial **1.1 dB** SI-SNRi and **1 dB** SDRi. In Appendix K, we randomly selected two long, uninterrupted audio samples from the VoxCeleb2 test set and mixed them together to generate sample outputs for RTFS-Net, AV-ConvTasNet, VisualVoice, AVLIT and the previous SOTA method, CTCNet. In Appendix L we studied additional downstream tasks to test the generalizability of RTFS-Net.

### 5.2 ABLATION STUDY

| AV Fusion Strategies | LRS2-2Mix | | RTFS-Net | RTFS-Net | Fusion | Fusion |
| | SI-SNRi | SDRi | Params (K) | MACs (G) | Params (K) | MACs (M) |
|---|---|---|---|---|---|---|
| CTCNet Fusion (adapted) | 11.3 | 11.7 | 528 | 14.3 | 197 | 6365 |
| CAF Block (ours) | **11.7** | **12.1** | **339** | **8.0** | **7** | **83** |

Table 2: Comparison of audio-visual fusion strategies on the LRS2-2Mix dataset. *RTFS-Net Parameters* and *MACs* indicate the entire network structure. The *Fusion Parameters* and *MACs* represent those used only during the fusion process.

**Cross-dimensional Attention Fusion (CAF)**. To demonstrate the value of our CAF Block, we used a reduced version of RTFS-Net-4 that limited the power of the separation network and emphasised the fusion network, see Appendix H.1. As a baseline, we substituted the CAF block with a TF-domain adaptation of CTCNet's fusion strategy. The previous SOTA method, CTCNet, uses a concatenation approach: it uses interpolation to upscale the video dimensions ($C_v \times T_v$) to the

dimensions of the audio ($C_a \times T_a$), concatenates along the channels and then uses a convolution to restore the audio channel dimension, $C_a$. Similar to Equation 8, we adapted this to a 3D TF-domain audio setting. In Table 2 we observed that our CAF Block outperformed the baseline by a large margin while reducing the number of parameters by **96.4%** and cutting the number of MACs by **98.7%**, suggesting multimodal fusion can be effectively performed with a well-designed, small network.

**Temporal-Frequency Attention Reconstruction (TF-AR)**. To test the efficacy of our RTFS Block's reconstruction method, i.e. the TF-AR units, we compared RTFS-Net-4 with a baseline that used interpolation and addition to perform the work of the TF-AR units, similar to the up-sampling process in U-Net (Ronneberger et al., 2015). We observe in Table 3 that our TF-AR units boosted performance by a substantial **1dB** in both performance metrics with an almost negligible affect on computational complexity due to the depth-wise convolutional nature of our TF-AR units.

| TF-AR | LRS2-2Mix SI-SNRi | SDRi | RTFS-Net Params (K) | RTFS-Net MACs (G) | TF-AR Params (K) | TF-AR MACs (M) |
|---|---|---|---|---|---|---|
| Without | 13.0 | 13.3 | **729** | **21.4** | **0** | **0** |
| With (ours) | **14.1** | **14.3** | 739 | 21.9 | 10 | 494 |

Table 3: Validation for the effectiveness of the TF-AR units on the LRS2-2Mix dataset. The *TF-AR Parameters* and *MACs* represent those used only in the TF-AR block.

**Spectral Source Separation** ($S^3$). In these experiments we further reduced the model configuration seen in Appendix H.1 by setting $C_a = 128$. We tested our $S^3$ Block against four alternative methods. *Regression* gives $\mathbf{a}_R$ directly to the decoder to decode into the target speaker's audio. *Mask* is the approach used by many SOTA AOSS methods such as Luo et al. (2020) and discussed in Section 3.3. *Mask + Gate* and *Mask + DW-Gate* both apply a convolutional gate, as seen in Luo et al. (2019), after the mask. The mask is fed into two convolutions with respective Tanh and Sigmoid activations. The outputs are multiplied together to form the final mask. *DW* here indicates the usage of depth-wise convolutions.

| Target Speaker Extraction Method | LRS2-2Mix SI-SNRi | SDRi | RTFS-Net Params (K) | RTFS-Net MACs (G) | Extraction Params (K) | Extraction MACs (M) |
|---|---|---|---|---|---|---|
| Regression | 10.0 | 9.9 | **208** | **3.0** | **0** | **0** |
| Mask | 10.8 | 11.2 | 224 | 3.6 | 16 | 534 |
| Mask + DW-Gate | 10.8 | 11.3 | 225 | 3.6 | 17 | 542 |
| Mask + Gate | 11.1 | 11.6 | 257 | 4.6 | 49 | 1595 |
| $S^3$ (ours) | **11.3** | **11.7** | 224 | 3.6 | 16 | 534 |

Table 4: Comparison of source separation methods on the LRS2-2Mix dataset. The *Extraction Parameters* and *MACs* represent those used only in the *Target Speaker Extraction Method*.

Table 4 shows the *Regression* approach was the least effective, with significantly lower metrics than the other methods. However, it also utilized the fewest MACs and parameters. For a very small increase in parameters, the *Mask* and *Mask+DW-Gate* approaches yielded far better results. A non depth-wise gate increased the performance further, but the number of MACs and parameters is significantly higher. However, all methods were eclipsed by the performance of our $S^3$ Block. This mathematics-based approach does not significantly increase the parameter or MAC count, thereby providing a performance enhancement to all TF-domain AVSS methods at no additional cost – likely benefiting AOSS methods as well. A visualization is available in Appendix E.

## 6 CONCLUSION

In this study we introduced RTFS-Net, a novel approach to AVSS that explicitly models time and frequency dimensions at a compressed subspace to improve performance and increase efficiency. Empirical evaluations across multiple datasets demonstrate the superiority of our approach. Notably, our method achieves remarkable performance improvements while maintaining a significantly reduced computational complexity and parameter count. This indicates that enhancing AVSS performance does not necessarily require larger models, but rather innovative and efficient architectures that better capture the intricate interplay between audio and visual modalities.

## REPRODUCIBILITY STATEMENT

The code for RTFS-Net was written in Python 3.10 using standard Python deep learning libraries, specifically PyTorch and PyTorch Lightning. In order to accommodate full reproducibility, we will open-source the code for RTFS-Net under the MIT licence on GitHub once this paper has been accepted into the conference. The code shall include all files used to reproduce the experiments seen in Table 1, including the conda environment we used to run the code, the full config files for RTFS-Net-4, RTFS-Net-6 and RTFS-Net-12, the weights for the pretrained video model and the RTFS-Net model code itself, including TDANet and all layers, blocks and networks mentioned in this paper. Datasets must be obtained separately from the references provided, see Appendix F, as they are the property of the respective publishers, but we provide the data-preprocessing scripts alongside this code-base. The GPU optimized PyTorch implementation of the SRU is already open-source and available on PyPi[2], supplied by the original author. Experimentation and training was accomplished using a single server with 8 NVIDIA 3080 GPUs, see Appendix I for additional details. For those who wish to recreate the code themselves, all hyperparameters are listed in Appendix H. Evaluation metrics and loss functions are described and mathematically defined in Appendix I.1 and G respectively, but we will additionally provide these in the code-base.

## ACKNOWLEDGEMENTS

This work was supported in part by the *National Key Research and Development Program of China* (No. 2021ZD0200301) and the *National Natural Science Foundation of China* (Nos. 62061136001, 61836014).

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

# A    Video Preprocessing Block

The VP Block is a 1D version of the RTFS Block, which is similar to the TDANet Block architecture (Li et al., 2023).

**Compression of time resolution**. Here, we change all 2D convolutions to 1D convolutions. As a result, we obtain the compressed global video features $\boldsymbol{V}_G$. This is written:

$$\boldsymbol{V}_G = \sum_{i=0}^{q-1} p(\boldsymbol{V}_i), \quad \boldsymbol{V}_G \in \mathbb{R}^{D \times \frac{T_\text{v}}{2^{q-1}}}. \tag{28}$$

**Dual-path architecture**. Since there is only one compressed dimension, we no longer need a dual path architecture. Following TDANet, we obtain

$$\bar{\boldsymbol{V}}_G = \text{Attn}(\boldsymbol{V}_G) + \boldsymbol{V}_G, \qquad \bar{\boldsymbol{V}}_G \in \mathbb{R}^{D \times \frac{T_\text{v}}{2^{q-1}}}. \tag{29}$$

where $\text{Attn}$ denotes the transformer used in TDANet, which consists of multi-head self-attention (MHSA) and a convolutional feed-forward network (FFN), with respective dropout and residual connections.

This attentional mechanism was first introduced to AOSS by DPTNet (Chen et al., 2020), but Li et al. (2023) found the subsequent bidirectional LSTM used in this implementation had negligible impact on performance, and could be replaced with a lightweight series of three 1D convolutions. See the paper and open-source code[3] for more details.

**Reconstruction of time and frequency resolution**. There is no frequency dimension, just the time dimension. We form 1D versions of the the TF-AR units by replacing the 2D convolutions and interpolations with 1D equivalents. The following reconstruction process is exactly the same, and so are all the equations in this section. $\bar{\boldsymbol{V}}_G$ is hence fused with each member of the multi-scale set, and then fused into a single feature map $\boldsymbol{V}_0''$.

# B    Sharing Parameters

| Shared Parameters | LRS2-2Mix | | Params | MACs |
|---|---|---|---|---|
| | SI-SNRi | SDRi | (K) | (G) |
| Not Shared | 13.6 | 13.9 | 2,183 | **21.9** |
| Shared | **13.7** | **14.0** | **739** | **21.9** |

Table 5: Effects of sharing parameters between the RTFS Blocks on the LRS2-2Mix dataset. We experimented with RTFS-Net-4.

In table 5 we consider the effects of sharing parameters between the RTFS Blocks, including the AP block. Sharing parameters means we instantiate a single RTFS Block instance, and use this single block as the AP Block and as all of the subsequent RTFS Blocks after the fusion. In other words, we take the output of the first RTFS Block, add the residual connection, and then pass that data to the exact same block again as an input. This means that all $R$ RTFS Blocks share the same weights and biases. Not sharing the parameters means that we instead instantiate $R$ separate RTFS Blocks, and iterate through them in turn.

Echoing the findings of Li et al. (2022), our study observed a small – but significant – increase in model performance, evidenced by the 0.1 dB increase in both SI-SNRi and SDRi. Additionally, sharing the parameters cuts the model size down from 2 million parameters down to just 739 thousand. The number of MACs is unchanged, as the number of calculations remains the same.

# C    Results for Various Compression Scales

In Table 6 we compared using different compression levels $q$ using the same RTFS-Net-4 configuration in Table 1. Increasing the compression factor results in a large drop in performance. However,

---

[3] https://github.com/JusperLee/TDANet

the reduction in computational complexity is correspondingly large. RTFS-Net with $q = 2$ and $q = 3$ both outperform the previous TF-domain SOTA method, VisualVoice (Gao & Grauman, 2021). As seen in Table 1, adding more layers (increasing $R$) will increase performance significantly, so using 12 or 16 layer configurations with a larger $q$ value would also be a valid option. However, we chose a lower $q$ and $R$ for simplicity. The RTFS-Net with $q = 1$ shows the result of applying TF-GridNet (Wang et al., 2023) directly to the AVSS setting. This model is powerful, and so with only 4 layers it nearly obtains the performance of our RTFS-Net-12 model in Table 1. However, the memory consumption is extremely large, and the model is extremely slow to train. A batch size of 1 is required for most standard GPUs. In addition, the purpose of this research is not to outperform the previous SOTA method, but to provide a lightweight and efficient solution to AVSS problem.

| $q$ | LRS2-2Mix SI-SNRi | SDRi | RTFS-Net Params (K) | RTFS-Net MACs (G) | Total Block Params (k) | Total Block MACs (M) | Training Memory (GB) |
|---|---|---|---|---|---|---|---|
| 1 | **14.7** | **14.9** | 747 | 60.5 | 489 | 55.9 | 5.74 |
| 2 | 14.1 | 14.3 | 739 | 21.9 | 481 | 17.3 | 3.99 |
| 3 | 12.8 | 13.1 | **737** | 12.4 | **479** | 7.8 | 3.44 |
| 4 | 11.5 | 12.0 | 740 | 10.2 | 482 | 5.5 | 3.33 |
| 5 | 11.0 | 11.4 | 746 | **9.7** | 487 | **5.0** | **3.29** |

Table 6: Comparison of different $q$ values (number of up-sampling steps) on the LRS2-2Mix dataset. The *Total Block Parameters* and *MACs* represent the parameters and MACs used in the $R = 4$ stacked RTFS Blocks.

## D  STRUCTURES FOR INDEPENDENTLY MODELING TIME AND FREQUENCY

In Table 7, we examine the effects of using different RNN structures as described in Equation 14. Each model employs the same RTFS-Net-4 configuration as presented in Table 1. We observe that both LSTM and SRU yield similar results when modeling the time and frequency dimensions. However, SRU demonstrates advantages over LSTM in terms of parameters, computational complexity, inference time, and training memory usage.

Additionally, we tested the effects of employing transformers. Specifically, we replaced the inter and intra SRUs with DPTNet (Chen et al., 2020) and adopted the transformer architecture from TDANet (Li et al., 2023) to avoid using heavy BiLSTM operations post-MHSA. The time-frequency domain transformer Wang et al. (2023) remained unchanged following the dual-path processing. In the table, this model is denoted as MHSA – it possesses fewer parameters and half the number of MACs, but shows an increase in training memory due to the transformer operations. Additionally, both the SI-SNRi and the SDRi metrics exhibited a 1dB decrease in performance, a significant drop, leading us to opt against this architecture despite its compact size.

We hypothesized that the unfolding before both SRUs might enhance transformer performance as well. However, this approach resulted in an extremely large model in all metrics: the training memory surged to 5GB, and the parameter count nearly reached 3 million. This configuration applies only the MHSA component of the transformer. Incorporating the FFN (MHSA-Unfold+FFN) led to an even larger and slower model. Despite these modifications, the performance remained substantially lower than that of the SRU configuration.

We can conclude that RTFS-Net's improvement in separation quality comes from the model design, not the choice of RNN. The SRU is implemented specifically to improve inference time and reduce the computational complexity.

## E  LOSS OF AMPLITUDE AND PHASE INFORMATION IN THE TRADITIONAL MASK APPROACH

In Figure 4 we examined the separation quality of two different target speaker extraction methods: our proposed $S^3$ Block and the traditional masking technique discussed in Section 3.3. For these results, we reused the $S^3$ and *Mask* models trained on the LRS2-2Mix dataset and shown in Table

| Structure Type | SI-SNRi | SDRi | Params (K) | MACs (G) | Time (ms) | Training Memory (GB) |
|---|---|---|---|---|---|---|
| RNN | 13.5 | 13.8 | 509 | 15.2 | 58.73 | **3.85** |
| GRU | 13.8 | 14.1 | 724 | 21.6 | 64.10 | 4.04 |
| LSTM | 14.0 | **14.3** | 832 | 24.8 | 61.92 | 4.08 |
| MHSA | 13.1 | 13.4 | **404** | **11.1** | - | 4.09 |
| MHSA-Unfold | 12.8 | 13.1 | 2,966 | 27.0 | - | 5.00 |
| MHSA-Unfold+FFN | 13.1 | 13.4 | 4,028 | 58.0 | - | 5.65 |
| SRU | **14.1** | **14.3** | 739 | 21.9 | 57.84 | 3.95 |

Table 7: Comparison of structures for modeling time and frequency on the LRS2-2Mix dataset. Here, *Time* denotes the time required for inference at 2s audio input. *Training Memory* denotes the GPU memory used for training with a batch size of 1.

4. We then selected three audio mixtures from the *VoxCeleb2-2Mix* dataset, creating an extremely challenging setting for the models, and applied the two models on each mixture.

In Figure 4a we can see that the traditional masking approach completely failed to capture the gold, vertical column in the middle of the expanded section. On the other hand, our $S^3$ not only captured this information, but also managed to preserve the striations seen in the original spectrogram.

In Figure 4b we see a similar result, our $S^3$ approach managed to accurately capture the gold stripes seen just right of the centre of the expanded section, whereas the the traditional masking approach completely excluded this part of the audio signal, resulting in a purple/black region in its place.

In Figure 4c we can see that our $S^3$ approach managed to accurately maintain the complex horizontal striations of the original audio, whereas the traditional masking approach produced a much more blurred result, with large purple regions showing it completely disregarded large chunks of important acoustic information.

The spectrograms in Figure 4, alongside the clear performance advantage seen in the evaluation metrics of Table 4, show the importance of respecting the underlying complex nature of the audio features obtained from the STFT. Failing to do so results in critical acoustic information being lost during the element-wise multiplication of the traditional masking approach, and hence the result obtained from the iSTFT is visibly degraded.

## F    DATASET DETAILS

We conducted our AVSS experiments on the LRS2 (Afouras et al., 2018a), LRS3 (Afouras et al., 2018b), and VoxCeleb2 (Chung et al., 2018) datasets. The different speakers are separated into different folders, allowing us to explicitly test speaker-independent performance. Additionally, we used a dataset partitioning consistent with existing AVSS methods to ensure fairness in obtaining results. For lip motion video acquisition, we used the FFmpeg tool[4] to change the sampling rate of all videos to 25 FPS. We then used an existing face extraction network (Zhang et al., 2016) to segment the lip region from the image and adjusted it to a $96 \times 96$ grayscale frame that contains only the image frames of the speaker's lip motion. For audio extraction, we similarly extracted the audio from the video using the FFmpeg tool and uniformly changed the sampling rate to 16 KHz. For comparison with existing AVSS methods, we used a mix of two speakers for training, but this dataset creation method can be used to create datasets for three or more speakers too.

The LRS2-2Mix (Afouras et al., 2018a) dataset is derived from BBC television broadcasts. The videos contain a variety of natural speakers and lighting conditions, and the audio contains background noise to bring it closer to real-world environments. Compared to previous lip-reading datasets (e.g. GRID (Cooke et al., 2006) and LRW (Son Chung et al., 2017)), LRS2-2Mix contains unrestricted vocabulary. It has full sentences rather than isolated words and shows greater diversity in terms of speakers and recording conditions. The dataset contains 11 hours of training, 3 hours of validation, and 1.5 hours of testing data.

---

[4]https://ffmpeg.org/

The LRS3-2Mix (Afouras et al., 2018b) dataset contains over 400 hours of video extracted from 5,594 TED and TEDx English-language talks downloaded from YouTube. TED talks are professionally produced and speakers typically use microphones, so it represents a relatively clean environment. The dataset includes 28 hours of training, 3 hours of validation and 1.5 hours of testing data.

The VoxCeleb2-2Mix (Chung et al., 2018) dataset was collected from YouTube videos containing naturally occurring noises such as laughter, cross-talk and different room acoustics. This provides a diverse and challenging set of speech samples for training the separation system. In addition, the faces in the videos differ in terms of pose, lighting, and image quality. This suggests that VoxCeleb2-2Mix provides a more challenging real-world environment. The dataset consists of 56 hours of training, 3 hours of validation and 1.5 hours of testing data.

## G   EVALUATION METRICS FOR AUDIO-VISUAL SPEECH-SEPARATION QUALITY

Following recent literature (Li et al., 2022), the scale-invariant signal-to-noise ratio improvement (SI-SNRi) and signal-to-noise ratio improvement (SDRi) were used to evaluate the quality of the separated speeches. These metrics were calculated based on the scale-invariant signal-to-noise ratio (SI-SNR) (Le Roux et al., 2019) and source-to-distortion ratio (SDR) (Vincent et al., 2006). For these evaluation metrics, a higher value indicates better performance. SI-SNRi and SDRi are defined as:

$$
\begin{aligned}
\text{SI-SNRi}(\boldsymbol{x},\ \boldsymbol{s},\ \hat{\boldsymbol{s}}) &= \text{SI-SNR}(\boldsymbol{s},\ \hat{\boldsymbol{s}}) - \text{SI-SNR}(\boldsymbol{s},\ \boldsymbol{x}), \\
\text{SDRi}(\boldsymbol{x},\ \boldsymbol{s},\ \hat{\boldsymbol{s}}) &= \text{SDR}(\boldsymbol{s},\ \hat{\boldsymbol{s}}) - \text{SDR}(\boldsymbol{s},\ \boldsymbol{x}).
\end{aligned}
\tag{30}
$$

## H   MODEL HYPERPARAMETERS

For all model versions, $R \in \{4, 6, 12\}$, we used the same hyperparamer settings.

**Encoder**. The STFT used a Hanning analysis window with a window size of 256 and a hop length of 128. The encoded feature dimension was $C_a = 256$.

**VP Block**. We used a TDANet block (Li et al., 2023) with a hidden dimension of 64, a kernel size of 3, an up-sampling depth of 4, replaced the gLNs with batch normalization and set the number of attention heads in the MHSA to 8. The feed forward network (FFN) had 128 channels. Model code was supplied by the original auther.

**AP and RTFS Blocks**. We used a hidden dimension of $D = 64$ as the reduced channel dimension for the RTFS Blocks. We chose $q = 2$ up-sampling layers in order to halve the length of the time and frequency dimensions, see Appendix C for different choices of $q$. The 4-layer bidirectional SRUs had a hidden dimension of $h_a = 32$, and an input dimension of $64 \times 8 = 256$ after unfolding. We used the 4-head attention configuration for the TF-domain attention (Wang et al., 2023), as defined in their paper. The model code for the TF-domain attention was taken from the ESPnet GitHub project[5], where it was published after the release of Wang et al. (2023)'s paper.

**CAF Block**. We used $h = 4$ attention heads.

### H.1   REDUCED MODEL

The reduced model used in Table 2 was achieved by using a modified version of the RTFS-Net-4 configuration defined above. Firstly, we set the hidden dimension of the RTFS Blocks to $D = 32$, replaced the multi-layer bidirectional SRUs with single layer unidirectional SRUs, and changed the kernel size and stride of the unfolding stage in Equation 13 to 4 and 2 respectively. This configuration is also used in Table 4, but with $C_a = 128$.

---

[5] `https://github.com/espnet/espnet.git`

# I    TRAINING HYPERPARAMETERS

For training we used a batch size of 4 and AdamW (Loshchilov & Hutter, 2018) optimization with a weight decay of $1 \times 10^{-1}$. The initial learning rate used was $1 \times 10^{-3}$, but this value was halved whenever the validation loss did not decrease for 5 epochs in a row. We used gradient clipping to limit the maximum $L_2$ norm of the gradient to 5. Training was left running for a maximum of 200 epochs, but early stopping was also applied. The loss function used for training was the SI-SNR (Le Roux et al., 2019) between the estimated $\hat{s}$ and original $s$ target speaker audio signals, see Appendix I.1.

## I.1    TRAINING OBJECTION FUNCTION

The training objective used is the scale-invariant source-to-noise ratio (SI-SNR) between the target speaker's original audio signal $s$ and the estimate returned by the model $\hat{s}$. The SI-SNR is defined as follows,

$$\text{SI-SNR}(s, \ \hat{s}) = 10 \log_{10} \left( \frac{||\boldsymbol{\omega} \cdot s||^2}{||\hat{s} - \boldsymbol{\omega} \cdot s||^2} \right), \quad \boldsymbol{\omega} := \frac{\hat{s}^T s}{s^T s}. \tag{31}$$

# J    INCREASING THE NUMBER OF LAYERS

Obtaining SOTA performance was not the focus of this paper. Rather, we aimed to produce a model with high efficiency while utilizing a small number of parameters. However, as an ablation study we explore higher values of $R$ in Table 8, including $R = 16$ and $R = 20$.

| Model | SI-SNRi | SDRi | Params (K) | MACs (G) |
|---|---|---|---|---|
| CTCNet 2022 | 14.3 | 14.6 | 7,043 | 176.2 |
| RTFS-Net-12 | 14.9 | 15.1 | **739** | **56.4** |
| RTFS-Net-16 | 15.2 | 15.5 | **739** | 73.7 |
| RTFS-Net-20 | **15.4** | **15.6** | **739** | 90.9 |

Table 8: Exploring higher $R$ values (number of RTFS Blocks, including the AP Block) on the LRS2-2Mix dataset.

From Table 8 we observed that increasing $R$ results in a significant increase in performance, with an associated increase in computational complexity. Note that since the RTFS Blocks share parameters, increasing $R$ does not increase the model size. RTFS-Net-20 outperforms CTCNet by over 1dB in both SDRi and SI-SNRi, while still utilizing only 91G MACs – just over half the MACs utilized by CTCNet. However, 20 RTFS Blocks requires a much larger GPU memory for training. As a result, NVIDIA 4090s were necessary in order to obtain these results.

# K    VISUALIZATION SAMPLES ON THE WEB PAGE

In this section, we aim to detail how we constructed the AVSS samples seen on our Web page[6]. To validate the generalizability of our approach, we purposefully chose videos from different scenarios and different genders with the aim of demonstrating the robustness of our model across a wide range of environments. We selected six males and two females to form four different mixed audio scenarios, including a variety of environments such as broadcasting studios, outdoors and variety shows.

We selected 8 videos from the VoxCeleb2 (Chung et al., 2018) test set, meaning none of the models saw these videos in their training or validation data. In order to extract the facial features of the target speaker, we used an existing face detection model (Zhang et al., 2016) as described in Appendix F. We then spliced the audios together in order to create the audio mixtures for the models, and placed the two original videos side by side for the demo.

---

[6]`https://anonymous.4open.science/w/RTFS-Net/AV-Model-Demo.html`

For each model, we passed in the mixed audio signal and the target speaker's facial features to obtain the target speaker's estimated separated audio stream for each of the 8 speakers. On our Web page, under each video is a link for each of the models tested. Upon opening the video, it will play the mixed audio in time to the video. Hovering the mouse over the desired speaker will switch the audio to the model's estimated separated audio stream for that speaker.

## L    GENERALIZING TO OTHER TASKS

In this section we explore how RTFS-Net can generalize to other tasks, namely: the presence of more speakers, and word error rate. These downstream tasks further test the robustness of our method.

### L.1    MORE SPEAKERS

Our model incorporates visual information from a single "target" speaker to isolate and extract the audio of this target speaker, hence we are not limited by the number of speakers present in the mixture. This integration of visual cues plays a crucial role in enhancing the accuracy of both our model and other AVSS models. We randomly sampled speakers from the LRS2-2Mix test set to obtain a dataset with three speakers. Adding more speakers increases the difficulty of the task, as from the perspective of extracting the target the speaker, the amount of background/extraneous noise has increased.

| Model | SI-SNRi | SDRi | Params (M) | MACs (G) |
|---|---|---|---|---|
| AV-ConvTasNet | 10.0 | 10.3 | 16.5 | - |
| AVLIT-8 | 10.4 | 10.8 | 5.8 | 36.4 |
| CTCNet | 12.5 | 12.8 | 7.0 | 167.2 |
| RTFS-Net-12 | 13.7 | 13.9 | 0.7 | 56.4 |

Table 9: Effects of training on the LRS2-2Mix dataset, and testing on the LRS2-3Mix dataset.

In Table 9 we observed that RTFS-Net-12 scaled much better to this harder task and outperformed CTCNet by a significant margin. While the presence of multiple speakers somewhat impacts the models' performance, it is important to note that these models were not trained on the dataset with three speakers, only tested on it.

### L.2    WORD ERROR RATE

In Table 10 we tested the speech recognition accuracy on the LRS2-2Mix test dataset. We utilized the publicly available Google Speech-to-Text API to obtain the recognition results. Our focus was on measuring the Word Error Rate (WER), where a lower rate is indicative of superior performance. The experimental results demonstrate that our model can achieve competitive speech recognition accuracy, validating the performance of our model in downstream tasks.

| Models | SDRi | WER (%) |
|---|---|---|
| Mixture | - | 84.91 |
| Ground-truth | - | 17.74 |
| CaffNet-C | 12.5 | 32.96 |
| Visualvoice | 11.8 | 34.45 |
| AV-ConvTasNet | 12.8 | 31.43 |
| AVLIT-8 | 13.1 | 31.85 |
| CTCNet | 14.6 | 24.82 |
| RTFS-Net-4 | 14.3 | 29.66 |
| RTFS-Net-8 | 14.8 | 27.42 |
| RTFS-Net-12 | 15.1 | 24.93 |

Table 10: Word Error Rate (WER) results on the LRS2-2Mix dataset.

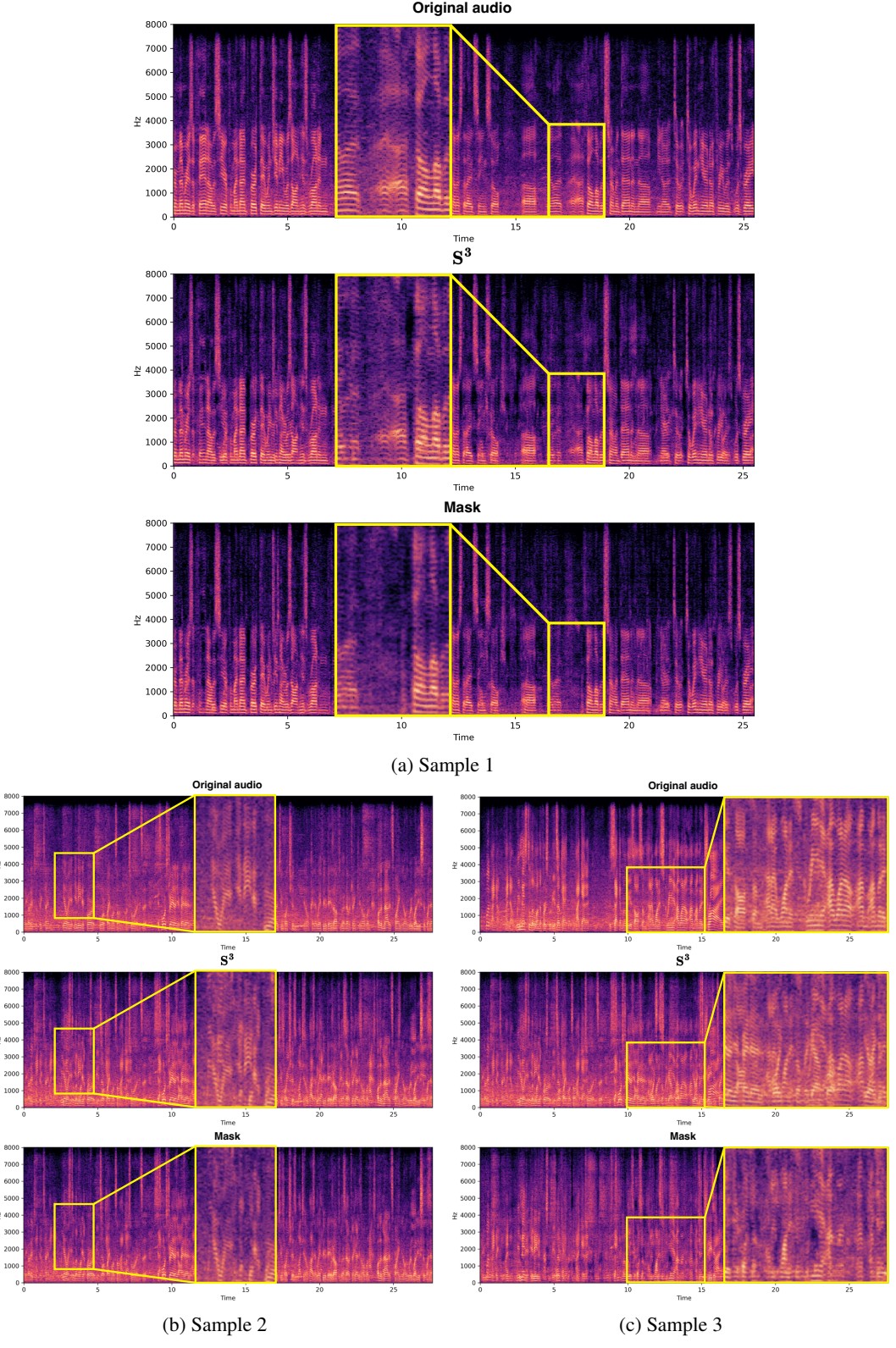

Figure 4: Comparison of target speaker extraction methods. In 4a, 4b and 4c, the top spectrogram is the target speakers original audio, the middle spectrogram is obtained using RTFS-Net-4 with $S^3$, and the bottom spectrogram is obtained using RTFS-Net-4 using the traditional masking approach.

