# OpenReview forum: "RTFS-Net: Recurrent Time-Frequency Modelling for Efficient Audio-Visual Speech Separation"
_ICLR.cc/2024/Conference — ICLR 2024 poster_

### Official Review · Reviewer_A4oN · 2023-10-31

**Soundness:** 4 excellent
**Presentation:** 3 good
**Contribution:** 4 excellent
**Rating:** 8
**Confidence:** 4

**Summary:**

The paper presents a novel Recurrent Time-Frequency Separation Network architecture that performs audio-visual source separation tasks effectively and efficiently. The model is characterized by three parts. First, each modality goes through its own processing module, and then the cross-dimensional attention fusion (CAF) consolidates information from both modalities. The spectral source separation block performs masking-based separation. The separation results show promising improvement given the compact size and computational efficiency the new model architecture introduces.

**Strengths:**

- The paper presents solid improvement compared to the existing baseline systems. Considering the amount of model compression the proposed model introduced, the improvement is significant.

- All the procedures and modules are well-defined with enough details.

- The choice of the model architectures makes sense, including the dual-path structure, attention-based consolidation, and complex masks.

- Ablation studies are thorough.

**Weaknesses:**

While the paper is packed with useful information, there are still some parts that need elaboration.

- As the authors mention, the dual-path RNN idea is not new to this problem. I understand that the authors chose SRU for their complexity-related considerations, but I also wonder if the audio processing module could benefit from its own self-attention mechanism, such as in the SepFormer model.

- The spectral source separation module might be the weakest contribution, because complex masks have been extensively studied in the audio-only source separation literature.

- I wish the paper provides more details on the TDANet block for video processing, which is relegated to the reference in the current version.

**Questions:**

- The authors chose to "add" f_1 and f_2 (eq 11) after the CAF processing. I think it's a little abrupt in the sense that there might be other choices that preserve the unique information that each vector learns, such as concatenation. Have the authors considered other ways to combine the two vectors?

---

> ### Author Response · Authors · 2023-11-19
> **Answer to Reviewer A4oN**
>
> We appreciate the effort you've invested in reviewing our manuscript and providing detailed feedback. Your insightful suggestions have been incredibly valuable, and we're confident that by addressing the points you've raised, we can enhance the quality of our paper. Below, we've addressed each of the concerns you've highlighted in your review:
>
> ***Q1: As the authors mention, the dual-path RNN idea is not new to this problem. I understand that the authors chose SRU for their complexity-related considerations, but I also wonder if the audio processing module could benefit from its own self-attention mechanism, such as in the SepFormer model.***
>
> **A1:** We experimented with different sequence processing architectures in Appendix D, which has now been updated to include several transformer designs. The self-attention mechanism leads to a decrease of 1 dB SI-SNRi, while significantly increasing the memory requirements of the model. Other transformer approaches resulted in extremely high parameter counts, which contradicts our aim of constructing a lightweight AVSS model.
>
> ***Q2: The spectral source separation module might be the weakest contribution, because complex masks have been extensively studied in the audio-only source separation literature.***
>
> **A2:** Some research has been conducted in the AOSS field using complex masks, but such studies are relatively scarce in the context of audio-visual speech separation. Additionally, to our knowledge, current state-of-the-art audio-only source separation methods, such as Sepformer, still employ real-valued masks. Indeed, TF-GridNet, the current AOSS SOTA method, does not use masks at all and directly computes the separated sources. We provided a study on all these different methods in Table 4. It is important to note that results in AOSS do not necessarily translate to the AVSS field.
>
> ***Q3: I wish the paper provides more details on the TDANet block for video processing, which is relegated to the reference in the current version.***
>
> **A3:** We have included an explanation of the video preprocessing block (VP BLOCK) in Appendix A.
>
> ***Q4: The authors chose to "add" $\mathbf{f}_1$ and $\mathbf{f}_2$ (eq 11) after the CAF processing. I think it's a little abrupt in the sense that there might be other choices that preserve the unique information that each vector learns, such as concatenation. Have the authors considered other ways to combine the two vectors?***
>
> **A4:** In Table 2 of our paper we compare the CAF block with pure concatenation (CTCNet Fusion (adapted)) of the audio and visual features. We observe that despite our CAF Block using only 3.6% ofthe parameters and 1.3% of the MACs, it outperformed pure concatenation by a large margin.
>
> The purpose for the CAF Block is to produce a low-parameter and MAC alternative to CTCNet’s fusion approach. If we directly concatenated $\mathbf{f}_1$ and $\mathbf{f}_2$, and then downscaled to the correct channel dimension using a convolution, this would be the same parameter and MAC count as the pure concatenation discussed above and in Table 2 of our paper. However, in order to fully answer your question, we ran an experiment where we concatenated $\mathbf{f}_1$ and $\mathbf{f}_2$ and then downscaled using a group convolution. Using a group convolution does not avoid the memory increase that comes from the concatenation, but it does avoid increasing the parameter and MAC counts significantly. The results show that this is an ineffective operation, and that a simple summation should be preferred.
>
> |  Method       |   Params  (K)  |  MACs  (M)  |  LRS2-2Mix SI-SNRi      |  LRS2-2Mix SDRi       |
> | --------------- | ------------- | --------------- | ------------- | ------------- |
> |  Add  (ours)  |  14.1       |  14.3  |  739  |  21,896  |
> |  Concat       |  13.7       |  13.9  |  741  |  21,913  |

---

> > ### Comment · Reviewer_A4oN · 2023-11-22
> > **Confirm**
> >
> > I confirm the author response was taken into account.

---

### Official Review · Reviewer_ZTD4 · 2023-11-01

**Soundness:** 3 good
**Presentation:** 2 fair
**Contribution:** 3 good
**Rating:** 6
**Confidence:** 4

**Summary:**

The authors build upon previous research in audio-only and audio-visual speech recognition by focusing on improving efficiency and fidelity of separated speech. They draw a lot of inspiration from the CTCNet paper and extend it to the TF domain to improve the efficiency. The solution has been evaluated on standard benchmark datasets and compared to previous state-of-the-art methods.

**Strengths:**

1. Audio samples of separation are available and source to be made available when the paper is published.
2. The writing is easy to follow.
3. Clear modeling details are provided.

**Weaknesses:**

1. The baseline methods listed in table 1 should include their references.
2. RTFS-Net-12 is only about 10% more efficient that CTCNet. How much difference does that make in practical applications?
3. Some of the comparison examples are not distinguishable to this reviewer. This makes me wonder how to interpret the relative SNR gains.

**Questions:**

1. Since the best performance is achieved with R=12, why not explore a higher R?
2. Have the authors considered conducting studies with human listeners? If the target application is ASR, would it be helpful to measure WER in a recognition task?

---

> ### Author Response · Authors · 2023-11-19
> **Answer to Reviewer ZTD4**
>
> Thank you very much for your valuable feedback. We have revised the manuscript and added some experiments to demonstrate the superiority of our model. We would also like to make some comments below to address your specific questions:
>
> ***Q1: The baseline methods listed in table 1 should include their references.***
>
> A1: Thank you for spotting that, we have updated the table.
>
> ***Q2: RTFS-Net-12 is only about 10% more efficient that CTCNet. How much difference does that make in practical applications?***
>
> A2: From the 10% figure we assume that the efficiency you mention is RTFS-Net's inference time of 109.9ms vs CTCNet's inference time of 122.7ms. The inference time is the time taken to process 2 seconds of audio. Scaling up to 1000 hours of audio, it would take 54.95 hours for RTFS-Net-12 and 61.35 hours for CTCNet. Since the separation performance of RTFS-Net is higher, and the time taken is shorter, the logical solution would be to use RTFS-Net as it would save around 6 and a half hours of GPU processing, and the associated electricity cost. Further, the performance drop from switching to RTFS-Net-4 is pretty minimal. Using this model, 1000 hours of audio would only take 28.9 hours.
>
> The total parameters and MACs are also important factors to be considered in practical applications. Compared to CTCNet, RTFS-Net-12 has reduced computational costs by three times, while the number of parameters utilized is only one-tenth.
>
> ***Q3: Some of the comparison examples are not distinguishable to this reviewer. This makes me wonder how to interpret the relative SNR gains.***
>
> A3: You are correct. The purpose of the demo, and indeed this paper, is to show that SOTA performance can be replicated using a fraction of the parameters and a greatly reduced model complexity. For example, in Appendix C we showed that if we do not compress the frequency and time resolutions of the audio features (i.e. q=1), we can outperform the SOTA method by a significant margin using only R=4 repetitions. Using 16 or more layers with q=1 would greatly increase the SNR further. However, the large memory requirement and computational complexity associated with this approach defeats the purpose of our paper.
>
> **Q4: Since the best performance is achieved with R=12, why not explore a higher R?**
>
> A4: A higher value of R leads to increased computational demands, which contradicts our intention to propose a lightweight model. Nonetheless, we still present a result as shown below. The results indicate that a higher R achieves better separation quality, but it comes with a more significant computational cost. Note that because we share parameters, increasing R only increases the MACs and not the parameter count.
>
> RTFS-Net with 16 and 20 layers on LRS2 dataset
>
> |  Model        |  SI-SNRi  |  SDRi  |  Params (K)  |  MACs (G)  |
> | --------------- | ----------- | -------- | -------------- | ------------ |
> |  RTFS-Net-12  |  14.9     |  15.1  |  739         |  56.4      |
> |  RTFS-Net-16  |  15.2     |  15.5  |  739         |  73.7      |
> |  RTFS-Net-20  |  15.4     |  15.6  |  739         |  90.9      |
>
> RTFS-Net-20 outperforms CTCNet by over 1dB in both SDRi and SI-SNRi, while still utilizing only 91G MACs – just over half the MACs utilized by CTCNet. However, 20 RTFS Blocks requires a much larger GPU memory for training. As a result, we had to use NVIDIA 3090/4090s in order to obtain these results in a short period of time.
>
> ***Q5: Have the authors considered conducting studies with human listeners? If the target application is ASR, would it be helpful to measure WER in a recognition task?***
>
> A5: We tested the Automatic Speech Recognition (ASR) results on the LRS2 test set. The results show that RTFS-Net generalizes well to downstream tasks, achieving almost identical performance to CTCNet – the previous SOTA method. For details, please refer to our response to the third question from Reviewer #1FXL.
>
> We have shared a demo in the supplementary material that can be listened to at your leisure, and we have shared our results with other members of our lab. However, conducting a formal study with large quantities of human participants or professional stenographers lies beyond the scope of our current research and is not feasible within the limited timeframe given for responding to reviews and questions.

---

> > ### Comment · Reviewer_ZTD4 · 2023-11-22
> > **Acknowledging authors' responses**
> >
> > Thank you for the thoughtful responses. Adding the new experimental results on ASR and the variation of R to the manuscript will be necessary to drive clarity. Additionally, the qualification on the subjective improvement is also important for the reader.
> >
> > Some of the tradeoffs have to be driven home more clearly.  If the biggest contribution is that the system achieves SOTA WER and human listening perception results at a tiny fraction of the computational cost, that has to be primary narrative with a clear focus on the importance of 10-13% savings in practical applications.

---

> > > ### Author Response · Authors · 2023-11-23
> > > **Answer to Reviewer ZTD4's response**
> > >
> > > Thank you for your valuable feedback and insightful suggestions regarding our submission. We appreciate the time and effort you have invested in reviewing our work and are eager to clarify some aspects based on your comments.
> > >
> > > 1.	The deadline for revisions has passed, but we understand and agree with your point about incorporating experimental results on Automatic Speech Recognition (ASR) and higher R values. We will include these results in the camera-ready version of our paper in a new appendix section, alongside a discussion.
> > > 2.	To effectively drive home the trade-offs, our manuscript presents a comprehensive range of objective evaluation metrics including SDRi, SI-SNRi, and PESQ, alongside the model size and computational complexity (parameters and MACs). These metrics are standardized, and the major metrics for speech-separation quality assessment in this field. Following previous works, we focused our analysis and discussion around these standard metrics to show that RTFS-Net-12 is 1/3rd the computational complexity and 1/10th the model size of CTCNet, the previous SOTA model in this domain. Additionally, RTFS-Net-12 outperforms CTCNet on all three metrics, over three standard benchmark datasets. This is the primary narrative of our paper, and our main contribution, not SOTA WER or human listening perception results, which are downstream tasks and outside the current scope of this study.
> > > 3.	In our main results table, the inference time is provided for reference. However, this metric can vary significantly depending on the hardware used, namely the CPU/GPU series and the communication between them. Usually, hardware-specific programming techniques are required to align the differences in MACs with the differences in inference time. We note that even without such techniques, the smaller versions of RTFS-Net can substantially reduce inference time, experiencing only a slight dip in performance in terms of SDRi and SI-SNRi. In fact, they reduce CTCNet’s inference time from 122.7ms to 57.8ms and 64.7ms, respectively, for R=4 and R=6. This represents a trade-off decision that is left to the discretion of the reader based on their specific application. Since the primary goal of our paper is to develop a lightweight and effective model, we focus on and prioritize small R values in our discussion and experimentation.
> > >
> > > We hope this clarification aligns with your expectations and provides a more comprehensive understanding of our work. We are grateful for the opportunity to discuss these aspects further and remain open to any additional feedback or suggestions you may have.

---

### Official Review · Reviewer_siqV · 2023-11-03

**Soundness:** 3 good
**Presentation:** 3 good
**Contribution:** 3 good
**Rating:** 8
**Confidence:** 4

**Summary:**

The paper proposes RTFS-Net, a new time-frequency (TF) domain audio-visual speech separation method. It introduces three main innovations:

* RTFS Blocks independently model time and frequency dimensions of audio
* Cross-dimensional Attention Fusion Block efficiently fuses audio and visual data
* Spectral Source Separation Block preserves phase/amplitude information

Experiments show RTFS-Net matches or beats prior time domain methods on LRS2, LRS3, and VoxCeleb2 datasets, while using 10x fewer parameters and 3-6x fewer computations.

RTFS-Net is the first TF model to surpass most contemporary time domain methods for audio-visual speech separation. It demonstrates TF domain methods can achieve good performance at lower computational cost through novel modeling of time-frequency spectrograms.

**Strengths:**

* Achieves near state-of-the-art performance for audio-visual speech separation while being very parameter and computationally efficient
* Outperforms all compared time domain methods, proving time-frequency domain modeling can achieve better performance if done effectively
* Innovative modeling of time and frequency dimensions independently in RTFS Blocks
* Attention-based fusion mechanism in CAF Block is very lightweight but fuses audio and visual data very effectively
* Spectral Source Separation Block properly handles phase/amplitude to avoid losing audio information
* Model code and weights will be released for full reproducibility

**Weaknesses:**

* Testing is limited to only 2 speaker mixtures. Performance with more speakers is uncertain.
* Missing PESQ evaluation in results which most other target speech extraction papers provide
* Doesn't include latest SOTA model comparison: Dual-Path Cross-Modal Attention for Better Audio-Visual Speech Extraction, ICASSP 2023.
https://arxiv.org/pdf/2207.04213.pdf. This gives superior performance for SI-SNRi and provides PESQ results as well. It does not provide MACs analysis.

**Questions:**

None

---

> ### Author Response · Authors · 2023-11-19
> **Answer to Reviewer siqV**
>
> We would like to thank you for your very positive feedback. We are honored that you have found potential in our work. Please find below our answers to the main questions that you have raised:
>
> ***Q1: Testing is limited to only 2 speaker mixtures. Performance with more speakers is uncertain.***
>
> **A1:**   We tested our model on two-speaker mixtures because most existing methods are verified in this manner. Furthermore, it is important to note that our model incorporates visual information from a single speaker to isolate and extract the audio of this target speaker. The integration of visual cues plays a crucial role in enhancing the accuracy of our model, as well as other AVSS models. Hence, we are not limited by the number of speakers.
>
> We randomly sampled speakers from the LRS2-2Mix test set to obtain a dataset with three speakers. The results indicate that the presence of multiple speakers somewhat impacts the model's performance. However, it is important to note that these models were not trained on the dataset with three speakers, only tested on the dataset with three speakers. In addition, RTFS-Net-12 outperformed CTCNet by a significant margin, scaling much better to the harder task.
>
> |  Model          |  SI-SNRi  |  SDRi  |
> | ----------------- | ----------- | -------- |
> |  AV-ConvTasNet  |  10.0     |  10.3  |
> |  AVLIT-8        |  10.4     |  10.8  |
> |  CTCNet         |  12.5     |  12.8  |
> |  RTFS-Net-12    |  13.7     |  13.9  |
>
> ***Q2: Missing PESQ evaluation in results which most other target speech extraction papers provide.***
>
> **A2:** We have added the PESQ evaluation metrics to Table 1 in the updated paper. As with SI-SNRi and SDRi, the PESQ scores show that RTFS-Net obtains comparable accuracy to the previous SOTA method (CTCNet) across all three datasets, while utilizing fewer parameters and MACs.
>
> ***Q3: Doesn't include latest SOTA model comparison: Dual-Path Cross-Modal Attention for Better Audio-Visual Speech Extraction, ICASSP 2023. https://arxiv.org/pdf/2207.04213.pdf. This gives superior performance for SI-SNRi and provides PESQ results as well. It does not provide MACs analysis.***
>
> **A3:** The model code for this ICASSP publication is not open source. Its results on the LRS3 dataset for the model “AV-TASNET” are different from ours, indicating that either their mixing method is different, or that their train/eval/test splits might be different. For similar reasons, we also did not include AV-Sepformer ([https://arxiv.org/abs/2306.14170](https://arxiv.org/abs/2306.14170)) in our paper. Both of these methods are Sepformer-like, meaning the parameter count for each model is at least 20 million. However, by comparing our paper with AV-Sepformer’s PESQ values, this method also does not outperform CTCNet or RTFS-Net (AV-Sepformer: 2.31, CTCNet: 3.00, RTFS-Net: 3.00).
>
> We would also like to emphasize that the purpose of RTFS-Net is not to obtain SOTA performance. As can now be seen in the updated Appendix C, we can easily outperform CTCNet by directly applying TF-GridNet to the audio-visual setting with only 4 RTFS-Net block layers. The focus of RTFS-Net is to obtain comparable performance to the SOTA method using a very small model footprint. Our contribution is a fast, efficient and quick-to-train model.

---

### Official Review · Reviewer_1FXL · 2023-11-08

**Soundness:** 4 excellent
**Presentation:** 4 excellent
**Contribution:** 3 good
**Rating:** 8
**Confidence:** 4

**Summary:**

In this paper, the authors present a novel time-frequency domain audio-visual speech separation method (Recurrent Time-Frequency Separation Network), a unique attention-based fusion technique for the efficient integration of audio and visual information, and a new mask separation approach. Results show that the proposed approach outperforms the previous SOTA method using only 10% of the parameters and 18% of the MACs. The authors claim that this is the first time-frequency domain audio-visual speech separation method to outperform all contemporary time-domain SOTA ones.

**Strengths:**

One main strength of this paper is the proposed Recurrent Time-Frequency Separation Network that processes the data in the frequency dimension, the time dimension, and the joint time-frequency dimension.

**Weaknesses:**

It is mentioned that the RTFS blocks share parameters (including the AP Block), leading to reduced model size and increased performance. Therefore, more description/explanation for this would be helpful.

**Questions:**

Are there any overlapping speech in the train/test data?
Do the authors perform any downstream task like speech recognition on the reconstructed speech?
One possible downstream task for speech separation is speech-to-speech dubbing. In this case, both the speech and background/nonspeech sound are needed. Have the authors looked into reconstructing background/nonspeech sound?

---

> ### Author Response · Authors · 2023-11-19
> **Answer to Reviewer 1FXL**
>
> We would like to thank you for the time taken to review our paper and for your extensive comments. Your suggestions are very helpful, and we believe we can significantly improve the paper by making the respective adjustments. We have compiled the issues that you have pointed out and answered them as follows:
>
> ***Q1: Explain the benefits of shared parameters.***
>
> **A1:** The purpose of sharing parameters between the RTFS Block layers is to create an overall network architecture that uses recurrent connections to continuously refine the clarity of the model’s output through the R “time steps”, just like an RNN. Additionally, this approach also reduces the quantity of model parameters, creating a more lightweight design. A more detailed discussion can be found in Appendix B on the updated paper. From the results in Appendix B (copied below), we can see that sharing the parameters reduces the model size from 2 million parameters to 739,000 parameters, while also delivering a slight performance improvement.
>
> | Shared Parameters | LRS2-2Mix SI-SNRi | LRS2-2Mix SDRi | Params (K) | MACs (G) |
> |-------------------|-------------------|----------------|------------|----------|
> | Not shared        | 13.6              | 13.9           | 2,183      | 21.9     |
> | Shared            | 13.7              | 14.0           | 739        | 21.9     |
>
> ***Q2: Is there any overlapping speech in the train/test data?***
>
> **A2:** In our approach, we did not impose any restrictions on the extent of overlap in the speech samples within the training and test sets. As a result, the data in both sets encompasses a diverse range of overlaps, including heavily overlapped, moderately overlapped, and non-overlapping speech. This variety ensures a comprehensive evaluation of the model's performance across different degrees of speech overlap.
>
> ***Q3:Downstream task.***
>
> **A3:** The speech separation task is a front-end task generally used to enhance speech recognition accuracy. To address your question, we tested the speech recognition accuracy on the LRS2-2Mix test dataset. We utilized the publicly available Google Speech-to-Text API to obtain the recognition results. Our focus was on measuring the Word Error Rate (WER), where a lower rate is indicative of superior performance. The experimental results demonstrate that our model can achieve competitive speech recognition accuracy, validating the performance of our model in downstream tasks. However, the task of separating background noise, also referred to as universal voice separation, was not investigated in this paper as it falls outside the scope of our current research. Our work, and all previous works that we compare with in our paper, are specifically designed for target speaker extraction – using the lip movements of a target speaker to extract that speaker’s audio signal. As there are no lip movements associated with background noise, implementing new modules, or modifying the architecture would be necessary in order to address the task of speech-to-speech dubbing. However, it represents an intriguing task for future research.
>
> |  Models         |  SDRi  |  WER (%)  |
> | ----------------- | -------- | ----------- |
> |  Mixture        |  -     |  84.91    |
> |  Ground-truth   |  -     |  17.74    |
> |  CaffNet-C      |  12.5  |  32.96    |
> |  Visualvoice    |  11.8  |  34.45    |
> |  AV-ConvTasNet  |  12.8  |  31.43    |
> |  AVLIT-8        |  13.1  |  31.85    |
> |  CTCNet         |  14.6  |  24.82    |
> |  RTFS-Net-4     |  14.3  |  29.66    |
> |  RTFS-Net-8     |  14.8  |  27.42    |
> |  RTFS-Net-12    |  15.1  |  24.93    |

---

> > ### Comment · Reviewer_1FXL · 2023-11-22
> >
> > It is great to see that the proposed approach works well with the downstream ASR task with reduced model size.

---

### Author Response · Authors · 2023-11-19
**Hope the reviewers will take note of our response**

Dear Reviewers,

After submitting the initial comments, we incorporated your feedback into a revised version of our paper, performed some additional experiments as you requested, and wrote a response to address your main concerns.

We hope to interact with you during the discussion and potentially further improve the quality of our paper.

Thank you very much in advance.

Kind regards,

Authors

---

### Meta-Review · Area_Chair_541n · 2023-11-30

**Metareview:**

This paper proposes a so-called Recurrent Time-Frequency Separation Network (RTFS-Net) for audio-visual based speech separation.   It consits of RTFS blocks, cross-dimensional attention fusion blocks and spectral source separation blocks. Extensive results are reported on numerous datasets with strong performance.  Most of the concerns raised by the reviewers have been cleared up by the authors in their rebuttal with detailed analysis and added experiments.  Overall, this is a strong paper.  There is a consensus among the reviewers to accept it.

**Justification For Why Not Higher Score:**

The strength of the paper is mainly on the strong SOTA performance of the proposed RTFS-Net. The novelty of the work is not overwhelmingly significant.

**Justification For Why Not Lower Score:**

The work is interesting. The performance is strong.  All reviewers are supportive of accepting the paper.

---

### Decision · Program_Chairs · 2024-01-16

Accept (poster)